# Towards Robust Benchmark of Object Hallucination on Multiple Images

## Abstract

Multimodal Large Language Models (MLLMs) are evolving into sophisticated agentic systems, engaging users in complex, multi-image scenarios. However, current MLLMs are limited by object hallucination, generating information inconsistent with visual evidence. Existing benchmarks, largely designed for single-image settings or offering only high-level multi-image assessments, fail to capture the nuanced causes of object hallucination, particularly under adversarial conditions. To address this, we introduce the Multi-Image Object Hallucination (MIOH) benchmark, a comprehensive framework specifically designed to diagnose MLLM vulnerabilities in complex multi-image contexts. MIOH integrates four object-centric tasks (existence, counting, attribute, position) with four controllable adversarial factors (visual context scale, perceptual difficulty, contextual bias, and misleading textual context). Through our systematic evaluation using MIOH, we reveal that even state-of-the-art models including GPT-5 and Gemini Pro still suffer from significant performance degradation under adversarial conditions, with models showing increased susceptibility to both false positive and false negative hallucinations when visual and linguistic contexts become challenging.

## 1 Introduction

With recent advances, Multimodal Large Language Models (MLLMs) are capable of reasoning over multiple images simultaneously. This task requires models not just to recognize content but to integrate and synthesize information from a diverse set of isolated visual inputs. Despite recent advancements, however, current MLLMs are limited by *object hallucination*, where models generate plausible but factually inconsistent descriptions about objects in the queried images.

Multi-image scenarios create new challenges for accurate object recognition, since the model has to simultaneously process increased visual complexity while creating coherent responses to the queries. That is, the multi-image context makes the model more susceptible to object hallucination, *e.g.*, incorrectly associating objects across different images or losing track of object identity. As object hallucination types and causes become increasingly complex in a multi-image setting, a more sophisticated approach is needed to assess model reliability.

In spite of the importance of object hallucination, no existing benchmarks have properly targeted the challenges of object hallucination under multi-image contexts. On one hand, most existing object hallucination benchmarks (*e.g.*, POPE (Li et al., 2023d), CHAIR (Rohrbach et al., 2018)) are limited to simple, single-image settings, which are hard to reveal the failure modes that emerge only in multi-image reasoning. Moreover, these benchmarks typically focus narrowly on existence and basic counting tasks, without extending to other object-related aspects such as attribute and spatial positioning that could provide more comprehensive evaluation of object hallucination. Also, their evaluation approaches are either overly simplistic binary questions or broad free-form captioning prompts like "describe this image", providing limited insights into specific recognition capabilities and being unable to effectively diagnose diverse hallucination patterns in MLLMs.

On the other hand, multi-image benchmarks like MMIU (Meng et al., 2024) and MuirBench (Wang et al., 2024a) are not designed to specifically assess robustness against object hallucination, failing to isolate and analyze scenarios where object hallucination is likely to be exacerbated in multi-image settings, *e.g.*, contextually ambiguous image sequences, small target objects, or misleading linguistic contexts.

To narrow the gap between these two worlds, we introduce the Multi-Image Object Hallucination (MIOH) benchmark, the first comprehensive evaluation framework specifically designed to assess object hallucination under multi-image settings. Unlike existing benchmarks, MIOH systematically integrates four foundational object-centric tasks (existence, counting, attribute, and position) with diverse question types adapted to multi-image scenarios, including comprehensive, comparative, and selective judgments, beyond simple binary questions. Furthermore, for more fine-grained diagnosis on when and why models fail, we introduce controllable adversarial factors, allowing us to isolate and analyze specific failure modes under multi-image contexts.

Our contributions are summarized as follows:

- We introduce MIOH, the first dedicated benchmark to systematically assess **object hallucination under multi-image contexts** with fine-grained diagnostic capabilities.
- We systemically define and integrate a set of **foundational object-centric tasks** curated for multi-image context, enabling thorough analysis beyond binary judgments.
- We design **adversarial scenarios** that potentially exacerbate hallucination in multiple images, largely unexplored in prior object hallucination benchmarks.

## 2 RELATED WORK

**Multimodal Large Language Models (MLLMs).** Following the success of LLMs, MLLMs have rapidly evolved through visual instruction tuning (Liu et al., 2023b), utilized by LLaVA (Liu et al., 2023b) and extended by InstructBLIP (Dai et al., 2023) and MiniGPT-4 (Zhu et al., 2023). Early MLLMs face challenges in cross-image reasoning due to limitations in visual token processing and inter-image semantic modeling (Li et al., 2023c; Dai et al., 2023; Cha et al., 2024; Alayrac et al., 2022). Recent work (Jiang et al., 2024; Li et al., 2024b; Laurençon et al., 2024; Yao et al., 2024; Lu et al., 2025; Deitke et al., 2025) has enabled multi-image understanding. Mantis (Jiang et al., 2024), LLaVA-NeXT-Interleave (Li et al., 2024b), and Idefics3 (Laurençon et al., 2024) leverage large-scale image-text data at training, and Qwen2.5-VL (Bai et al., 2025), InternVL3.5 (Wang et al., 2025), and Gemini-2.5-Pro (Comanici et al., 2025) demonstrate powerful cross-image reasoning capabilities.

**Object Hallucination in MLLMs.** Object hallucination, defined as MLLMs generating plausible but inaccurate object descriptions inconsistent with visual inputs, remains a critical challenge (Rohrbach et al., 2018; Dai et al., 2022). Systematic analysis has pinpointed causes across the MLLM pipeline: data-related issues such as annotation noise (Liu et al., 2023b), limitations of vision encoder in fine-grained semantics (Zhai et al., 2023), insufficient modality alignment (Liu et al., 2024a), and inherited LLM biases such as weak context attention (Wang et al., 2024c). Recent studies reveal additional visual vulnerabilities, *e.g.*, perception of small or occluded objects (Zhang et al., 2024a), contextual bias due to object co-occurrence patterns (Li et al., 2023d) and semantic similarities (Li et al., 2024a). MLLMs are also linguistically susceptible to sycophantic alignment with user beliefs and context hijacking from misleading narratives (Zhao et al., 2024; Mehrotra et al., 2024).

Mitigation strategies, *e.g.*, data augmentation (Sarkar et al., 2024), preference optimization (Zhang et al., 2024b), and inference-time interventions (Zhao et al., 2025; He et al., 2025), have primarily targeted single image scenarios. Multi-image contexts amplify hallucination challenges, requiring models not only to recognize objects accurately, but to track them and maintain contextual consistency across distinct images (Wu et al., 2024). This increased complexity requires a new benchmark tailored to assess object hallucination on multiple images, which is the focus of our work.

**Benchmarks for MLLMs.** Early MLLM benchmarks focus on single-image scenarios across various tasks including visual question answering, reasoning, and compositional understanding (Goyal et al., 2017; Fu et al., 2023; Li et al., 2023a; Liu et al., 2023a; 2024b). Recently, several benchmarks (Wang et al., 2024a; Meng et al., 2024; Liu et al., 2024c; Jiang et al., 2024; Fu et al., 2024; Li et al., 2023b) assess general reasoning capabilities across multiple images, though none of them specifically target object hallucination.

In parallel, object hallucination has been addressed through specialized benchmarks, including discriminative approaches using binary questions (Li et al., 2023d; Guan et al., 2024) and generative approaches directly assessing free-form descriptions (Rohrbach et al., 2018; Kaul et al., 2024b; Sun et al., 2023; Wang et al., 2023). They typically focus on existence and basic counting tasks, with limited coverage of attributes or spatial relations. Also, they predominantly employ simple binary

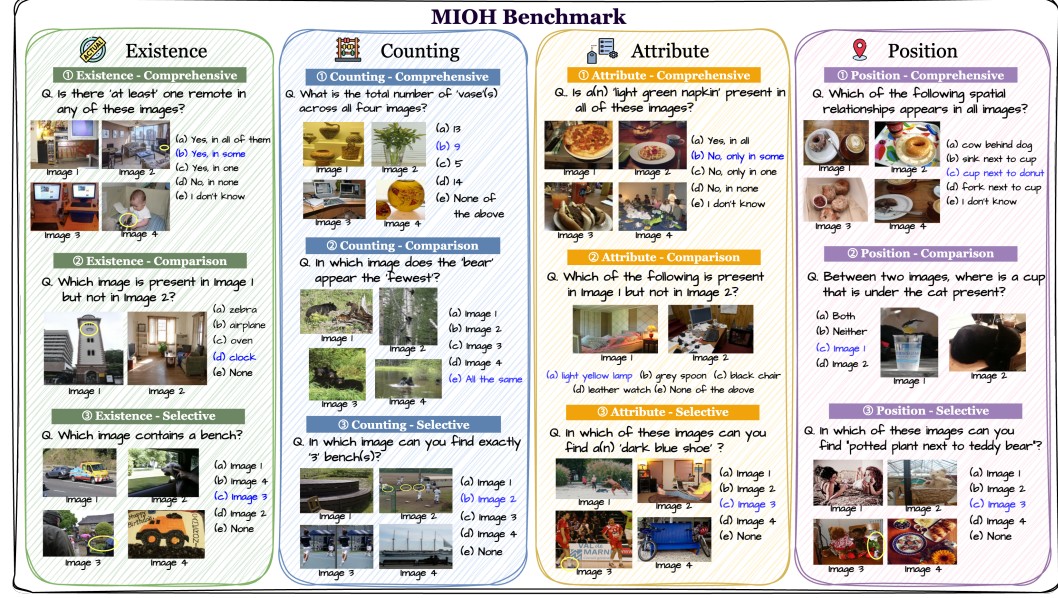

Figure 1: **Overview of our MIOH Benchmark.** MIOH evaluates object hallucination in multi-image contexts across four core tasks: existence, counting, attribute, and position. Each task includes three question types (comprehensive, comparative, and selective) designed to probe different aspects of multi-image reasoning capabilities.

questions or captioning tasks, confined to single-image settings. Consequently, no existing frameworks systematically evaluate object hallucination under multi-image contexts, leaving a critical gap in assessing model reliability across complex scenarios.

## 3 OVERALL DESIGN OF OUR MIOH BENCHMARK

We introduce the Multi-Image Object Hallucination (MIOH) benchmark, designed to systematically evaluate object hallucination in MLLMs under multi-image contexts. MIOH is designed with two tiers, one for measuring foundational capabilities of MLLM with general cases (Sec. 3.1), and another to scrutinize the model's ability to deal with particularly challenging examples (Sec. 3.2).

### 3.1 BENCHMARK FOR FOUNDATIONAL CAPABILITIES

We first define four core object-centric tasks that represent fundamental visual understanding capabilities: existence, counting, attribute, and position. These tasks collectively cover the primary tasks in existing object hallucination benchmarks (Rohrbach et al., 2018; Li et al., 2023d; Lovenia et al., 2023; Kaul et al., 2024a; Wang et al., 2024b; Chen et al., 2024) and serve as the foundation for assessing object-centric capabilities of MLLMs (Fu et al., 2023; Liu et al., 2024b; Jing et al., 2023; Sun et al., 2023; Qiu et al., 2024; Wang et al., 2024a; Villa et al., 2025).

In order to design our benchmark distinguished from single-image settings, it is important to require MLLM models to perform deep reasoning on the multiple input images. For this, we design three types of questions for more multifaceted evaluation, beyond simple binary ones: *comprehensive* (collectively understanding information across all images), *comparative* (identifying differences between images), and *selective* (retrieving a particular image described by the question) abilities. Fig. 1 illustrates questions in each category.

**Existence.** Verifying the presence or absence of a particular object is a fundamental test for object hallucination, but existing benchmarks (Rohrbach et al., 2018; Li et al., 2023d; Lovenia et al., 2023; Chen et al., 2024) have centered this task on a single-image (Bai et al., 2024). MIOH generalizes this to multi-image scenarios, incorporating three types of questions: comprehensive questions to verify presence of an object across the entire image set (*e.g.*, in all or any images), comparative questions focusing on differences between specific images (*e.g.*, in Image 1 but not in Image 2), and selective questions (*e.g.*, in which image) to identify a specific image where an object appears.

**Counting.** Beyond mere existence, a precise enumeration of a particular object is a common failure point of MLLMs (Zhang et al., 2024a; Gunjal et al., 2024; Jing et al., 2023; Tamarapalli et al., 2025). Similarly to the above, comprehensive questions ask to collectively count across all images. Comparative questions ask which image has the most or fewest instances, and selective questions ask to find an image that contains an exact number of objects.

**Attribute.** This task is to assess the model's more detailed and compositional understanding of the scenes, requiring to bind visual properties (*e.g.*, color, texture) with objects; *e.g.*, distinguishing a 'red car' from just a 'car'. Some benchmarks have touched attribute hallucination (Wang et al., 2023; Kaul et al., 2024b; Bai et al., 2024), but we fully integrate it as one of the fundamental tasks, on par with Existence and Counting. Again, we design three types of questions: verifying the presence of a specific combination of a particular attribute-object pair across the image set (comprehensive), distinguishing attributes across images (comparative), and finding a particular image containing the object with the correct attribute (selective).

**Position.** The last task presents the most complex challenge, evaluating a model's understanding of spatial relationships between two objects (*e.g.*, 'a doughnut in front of a dog'). Like attributes, spatial relations introduce compositional complexity and demand accurate scene parsing beyond object presence. Comprehensive questions test the consistency of a spatial relationship across all images, while comparative questions identify changes in this relationship across images. Selective questions ask a model to identify a specific image where the scene is depicted.

## 3.2 ADVERSARIAL PRESSURES AND DATASET CURATION

In addition to the standard questions covering general queries for MLLMs, we present another benchmark containing a significantly more challenging set of tasks, intentionally designed with adversarial pressure. In order to systematically probe the model's vulnerabilities under particularly challenging but still realistic conditions and to eventually diagnose the root causes of object hallucination, MIOH features common visual and linguistic misleading factors detailed below. The visual factors stem from challenges inherent to perceptual and contextual difficulties, while the lingual factor tests the model's robustness against distracting textual contexts. A core premise of our design is that MLLMs should be able to ground their answers in the provided images; therefore, we explicitly prompt the model to prioritize the visual content, even when the accompanying text provides subtly misleading or distracting information.

**Visually Challenging Factors.** First, inspired by findings that MLLMs struggle to identify information across a large set of images (*i.e.*, 'Visual Haystack' problem) (Wu et al., 2024), we evaluate the impact of **visual context scale** on object hallucination. Specifically, we vary the number of input images for the same question and measure the performance degradation as the demand for robust information integration exponentially increases.

Second, small or partially occluded objects tend to be harder to detect due to their low resolution and lack of information (Zhang et al., 2024a; Liu et al., 2025; Wei et al., 2025). In addition to these **perceptually challenging examples** (even to humans), we also curate samples that image encoders pre-trained on large-scale datasets particularly suffer to detect, even if they might look obvious to human eyes. They are for some reason hard to detect by a machine learning model, so testing the MLLM performance on these images would be meaningful to gauge its robustness.

As the opposite case to the above, **strong contextual bias** might mislead the model to believe presence of a particular object (Datta & Sundararaman, 2025; Li et al., 2023d; 2024a). For instance, from a scene of a kitchen with lots of typical kitchenware, a visual encoder might score high for a frying pan which does not exist there, due to the imperfect visual encoder. To measure the robustness of the MLLMs in such situation, we curate distracting samples exploiting contextual bias using co-occurrence statistics and similarities measured by pre-trained encoders.

**Lingual Challenging Factors.** We aim to analyze how a misleading textual context could affect visual judgment, even when the visual evidence is clear. This approach is grounded in the established understanding that an interactive MLLM's visual grounding can be compromised by prior context or the user prompt (Park et al., 2024; Qiu et al., 2024; Cao et al., 2024; Lin et al., 2024). We analyze this vulnerability based on two aspects of linguistic pressure: whether the model tends to align with a user's subjective and uncertain belief (**sycophantic pressure**) (Zhao et al., 2024), and

Figure 2: **MIOH Benchmark Construction Pipeline.**

whether the model tends to obey a direct counterfactual assertion that contradicts with the visual facts (**instructional override**) (Qiang et al., 2023; Mehrotra et al., 2024).

## 4 DETAILED BENCHMARK CONSTRUCTION

In this section, we provide details how we curate data samples (Sec. 4.1), how we generate questions (Sec. 4.2), and how we construct the adversarial benchmark (Sec. 4.3).

### 4.1 DATASET CURATION

Considering the importance of annotation quality in object hallucination benchmarks, we carefully select three datasets that provide high-quality annotations for different object-centric tasks, addressing critical limitations in existing datasets. We utilize **COCO-ReM** (Singh et al., 2024) for existence and counting tasks to leverage its systematic re-annotation that addresses the original COCO's incomplete object masks, missing instances, and inaccurate bounding boxes. For attributes, we use **PACO** (Ramanathan et al., 2023), providing standardized attribute labels across diverse object categories. For spatial relations, we use **SVG** (Park et al., 2025), which offers positional relationship annotations. Unlike existing scene graph datasets such as Visual Genome (Krishna et al., 2017) and GQA (Hudson & Manning, 2019), which typically provide only 1.5 relationships per subject due to human labeling constraints and insufficient coverage of spatial relationships, SVG more completely annotates scene-level relationship.

From this collection of datasets, we generate positive and negative samples for each task defined in Sec. 3.1. For Existence task, each question consists of an (IMAGE, OBJECT) pair, and it is labeled as positive if the IMAGE contains the OBJECT, while negative otherwise. For Counting, an example consists of (IMAGE, OBJECT, COUNT), and it is considered positive if the IMAGE presents the target OBJECT exact number of COUNT times. The Attribute and Position tasks are similarly constructed with an example of (IMAGE, ATTRIBUTE, OBJECT) and of (IMAGE, OBJECT1, RELATION, OBJECT2), respectively, and we label them according to the correctness based on the ground-truth annotations in the original datasets. In this way, we have a pool of positive and negative samples per each task, which will be utilized to generate multi-image questions.

### 4.2 QUESTION GENERATION

Using the prepared positive and negative samples per task, we generate multi-image questions that instantiate the three types of questions (comprehensive, comparative, and selective) across multi-image contexts for each of the four core tasks. As all ingredients are ready, this step can be easily automated by rule-based templates; *e.g.*, for Counting, we construct a question by randomly selecting $N$ IMAGEs containing a particular OBJECT, and the sum of COUNT per each example is the right answer. Other tasks follow a similar procedure to generate the question and correct answer. All questions are constructed in multiple-choice (MCQ) format by incorporating incorrect answers. We also include the "None of the above" options to more precisely assess the model's understanding. Fig. 1 illustrates actual questions across all tasks and question types, and we provide the question templates in Appendix C.

### 4.3 ADVERSARIAL BENCHMARK CONSTRUCTION

To create a particularly challenging benchmark, we additionally consider the level of difficulty of each primitive example (Sec. 4.1). We construct each adversarial pressure in Sec. 3.2 as follows.

**Visually Challenging Factors.** For the *visual context scale*, we vary the number of images among $\{2, 4, 8, 10\}$. For the *perceptual difficulty*, we collect **hard positive (HP)** (IMAGE, OBJECT) pairs such that the OBJECT is present but difficult to detect using the following two approaches: 1) Rule-based filtering, to collect (IMAGE, OBJECT) pairs with small or heavily occluded bounding box or segmentation mask for the OBJECT, and 2) CLIP-based semantic filtering, to select (IMAGE, OB-JECT) pairs with abnormally low CLIP similarity between the IMAGE a text prompt "A photo of OBJECT". To implement the *contextual bias*, we take a statistical approach by collecting **hard negatives (HN)** (IMAGE, OBJECT) pairs where the target OBJECT is absent but contextual cues mis-leadingly suggest its presence. Specifically, we estimate the co-occurrence probability between all object-pairs from the training data. We then collect (IMAGE, OBJECT) pairs that contain frequently co-occurring objects but not the target OBJECT itself. We additionally apply CLIP-based semantic confusion by identifying images that show high similarity with target prompts despite lacking the actual object, creating scenarios where visual-text misalignment leads to false positive predictions.

When we generate the questions as described in Sec. 4.2, we can gradually include these challenging examples; for instance, when we create a question with $N$ images, 0 to $N$ challenging example may be included in the final question, making it more challenging when it gets close to $N$.

**Lingual Challenging Factors.** For linguistic adversarial pressure, we prepend misleading textual context to the question prompt. For *sycophantic pressure*, we use uncertain, personal tones express-ing user beliefs that conflict with visual evidence (*e.g.*, "I looked carefully but don't see what I'm looking for"). For *instructional override*, we employ authoritative, declarative statements that di-rectly contradict visual facts (*e.g.*, "Analysis confirms the target is not in these images"). These contexts are strategically applied based on ground truth patterns to create visual-linguistic conflicts, while maintaining the core instruction to prioritize visual evidence.

**Quality Assurance and Validation.** Despite using high-quality re-annotated datasets and system-atic filtering, we have manually validated annotation errors and conceptual ambiguities. Each ques-tion undergoes review by three independent reviewers, with majority vote resolution ensuring the final benchmark's reliability for MLLM hallucination assessment. After validation, the benchmark consists of 20,518 questions across 82,412 images.

## 5 BENCHMARK RESULTS AND DISCUSSION

We conduct a comprehensive comparative study using our MIOH over the state-of-the-art MLLMs, including GPT-5 (OpenAI, 2025) and Gemini-2.5-Pro (Comanici et al., 2025). Among open-source models, we choose the LLaVA (Li et al., 2024b) series, the Qwen (Bai et al., 2025) series, InternVL (Wang et al., 2025), Phi-4-multimodal (Abouelenin et al., 2025), MiniCPM-V (Yao et al., 2024), Ovis-2.5 (Lu et al., 2025), and Mantis-8B (Jiang et al., 2024). For reproducibility, the decoding temperature is set to 0 for all experiments. All experiments were conducted on four NVIDIA A6000 GPUs.

### 5.1 OVERALL PERFORMANCE

We first present overall performance results demonstrating the extent of object hallucination chal-lenges across different model categories and tasks. Tab. 1 reports the comprehensive evaluation results over 30 models, revealing that multi-image object hallucination remains as significant chal-lenge, with an overall average accuracy of only **37.0%**. Not surprisingly, a clear performance gap is measured between proprietary and open-source models. The leading models, Gemini-2.5 Pro (60.9%) and GPT-5 (58.1%), set the state-of-the-art, but are still far from perfect. Top-performing open-source models, such as Qwen2.5-VL-7B (48.2%) and MiniCPM-V-2.6 (46.9%), demonstrate strong capabilities but lag behind the frontier models.

### 5.2 TASK-SPECIFIC VULNERABILITY

We conduct detailed analyses to identify specific failure patterns, characterize the impact of dif-ferent adversarial pressures, and examine how visual-linguistic conflicts affect model behavior in

| Model | Existence | | | | | | | Counting | | | | | | |
|---|---|---|---|---|---|---|---|---|---|---|---|---|---|---|
| | Easy | HN | HP | NI | LC S | LC I | Avg | Easy | HN | HP | NI | LC S | LC I | Avg |
| **Overall** | 64.1 | 58.5 | 57.2 | 29.3 | 38.7 | 36.5 | **49.1** | 29.8 | 29.1 | 26.3 | 18.9 | 29.5 | 16.8 | **25.4** |
| **GPT-5** | 90.7 | 81.9 | 77.3 | 55.3 | 88.0 | 44.9 | **73.5** | 53.7 | 51.9 | 39.8 | 35.6 | 57.4 | 6.3 | **40.3** |
| **Gemini-2.5 Pro** | 80.8 | 80.1 | 80.2 | 56.1 | 88.6 | 34.7 | **71.4** | 60.6 | 57.7 | 42.8 | 39.2 | 61.2 | 4.3 | **43.7** |
| **Qwen2-VL-2B** | 73.9 | 71.7 | 69.2 | 32.5 | 69.9 | 30.5 | **60.2** | 33.0 | 32.3 | 21.0 | 21.4 | 26.3 | 17.1 | **26.0** |
| **Qwen2.5-VL-3B** | 79.8 | 75.1 | 73.4 | 44.5 | 38.7 | 42.9 | **61.3** | 36.5 | 36.1 | 22.4 | 17.0 | 26.3 | 18.2 | **26.7** |
| **Qwen2-VL-7B** | 85.1 | 77.4 | 73.9 | 30.4 | 25.9 | 43.8 | **59.2** | 42.6 | 41.8 | 23.6 | 17.5 | 36.2 | 16.4 | **30.0** |
| **Qwen2.5-VL-7B** | 84.6 | 75.9 | 71.6 | 28.0 | 83.6 | 50.5 | **67.4** | 40.1 | 38.3 | 23.6 | 21.3 | 35.0 | 19.1 | **29.5** |
| **LLaVA-v1.6 (Mistral-7B)** | 37.4 | 67.1 | 41.2 | - | 55.4 | 28.1 | **45.2** | 26.7 | 23.8 | 26.7 | - | 27.4 | 13.7 | **24.3** |
| **LLaVA-Interleave (Qwen-0.5B)** | 40.1 | 38.8 | 33.8 | 23.0 | 30.4 | 56.7 | **37.5** | 25.5 | 28.9 | 23.8 | 16.4 | 24.3 | 20.7 | **23.5** |
| **LLaVA-Interleave (Qwen-7B)** | 41.5 | 48.1 | 73.7 | 32.1 | 34.4 | 23.0 | **44.8** | 30.3 | 30.1 | 29.4 | 20.6 | 31.0 | 21.7 | **27.4** |
| **LLaVA-Interleave (Qwen-7B-DPO)** | 64.6 | 47.9 | 58.7 | 31.4 | 48.7 | 43.8 | **53.3** | 31.7 | 33.5 | 30.1 | 18.4 | 29.2 | 22.2 | **27.7** |
| **LLaVA-OneVision (Qwen2-0.5B-SI)** | 37.8 | 42.6 | 36.6 | 30.9 | 53.9 | 56.0 | **42.9** | 24.0 | 22.2 | 24.0 | 18.5 | 25.4 | 23.1 | **22.8** |
| **LLaVA-OneVision (Qwen2-0.5B-OV)** | 65.4 | 27.9 | 31.6 | 20.2 | 66.7 | 57.4 | **47.8** | 26.4 | 28.9 | 24.2 | 14.3 | 27.3 | 23.4 | **24.2** |
| **LLaVA-OneVision (Qwen2-7B-SI)** | 73.4 | 69.3 | 44.1 | 31.2 | 67.3 | 29.0 | **51.7** | 30.6 | 27.7 | 30.5 | 18.4 | 27.0 | 10.2 | **24.8** |
| **LLaVA-OneVision (Qwen2-7B-OV)** | 82.5 | 75.7 | 58.6 | 27.7 | 26.3 | 22.2 | **48.8** | 29.0 | 25.1 | 27.6 | 19.8 | 23.3 | 17.6 | **23.8** |
| **LLaVA-OneVision (Qwen2-7B-OV-Chat)** | 83.0 | 75.9 | 46.1 | 27.3 | 33.8 | 28.4 | **49.2** | 30.6 | 26.1 | 26.9 | 20.9 | 23.4 | 17.5 | **24.3** |
| **InternVL3.5-1B** | 74.1 | 68.5 | 67.7 | 46.4 | 43.4 | 36.0 | **57.3** | 29.0 | 31.2 | 24.9 | 18.0 | 32.3 | 19.2 | **25.8** |
| **InternVL3.5-2B** | 52.6 | 48.6 | 48.3 | 23.0 | 44.9 | 42.5 | **44.4** | 22.4 | 23.7 | 20.9 | 17.2 | 22.4 | 16.7 | **20.3** |
| **InternVL3.5-4B** | 58.9 | 76.1 | 62.7 | 20.8 | 46.4 | 27.6 | **49.4** | 27.7 | 30.4 | 26.8 | 16.8 | 28.0 | 16.4 | **24.7** |
| **InternVL3.5-8B Pretrained** | 67.9 | 63.2 | 61.0 | 24.3 | 59.4 | 29.4 | **51.8** | 30.7 | 28.3 | 28.0 | 17.1 | 25.8 | 9.8 | **23.5** |
| **InternVL3.5-8B Instruct** | 68.4 | 61.4 | 73.4 | 32.2 | 63.3 | 37.3 | **55.9** | 33.3 | 31.9 | 28.9 | 15.4 | 35.8 | 13.6 | **26.4** |
| **InternVL3.5-8B MPO** | 68.5 | 61.9 | 74.6 | 20.2 | 34.4 | 24.9 | **48.6** | 33.8 | 31.4 | 29.5 | 15.3 | 26.1 | 15.9 | **25.4** |
| **InternVL3.5-8B** | 70.8 | 64.7 | 64.1 | 20.8 | 67.1 | 42.9 | **54.1** | 27.0 | 28.6 | 30.3 | 14.3 | 39.2 | 17.1 | **26.3** |
| **Mantis-8B (CLIP-Llama3)** | 47.5 | 56.2 | 41.4 | 24.2 | 39.0 | 29.1 | **40.3** | 24.1 | 25.2 | 24.3 | 20.3 | 23.9 | 18.0 | **22.5** |
| **Mantis-8B (SIGLIP-Llama3)** | 48.9 | 45.4 | 42.5 | 27.8 | 40.2 | 29.5 | **40.1** | 27.3 | 27.7 | 27.1 | 18.5 | 28.7 | 20.5 | **25.0** |
| **MiniCPM-Llama3-V-2.5** | 43.3 | 43.6 | 38.8 | 9.0 | 54.3 | 26.8 | **37.4** | 7.5 | 7.8 | 8.6 | 3.8 | 12.4 | 6.6 | **8.0** |
| **MiniCPM-V-2.6** | 84.1 | 76.8 | 73.6 | 44.8 | 46.5 | 43.1 | **63.1** | 32.5 | 31.9 | 25.3 | 15.7 | 31.5 | 18.9 | **25.6** |
| **Ovis2.5-2B** | 73.4 | 20.4 | 43.3 | 31.2 | 40.3 | 32.9 | **39.3** | 28.6 | 22.4 | 22.0 | 22.2 | 20.4 | 25.7 | **23.0** |
| **Ovis2.5-9B** | 80.1 | 21.6 | 46.3 | 31.3 | 43.6 | 43.6 | **43.2** | 32.0 | 23.5 | 35.1 | 23.0 | 27.8 | 27.8 | **29.0** |
| **Phi-4-multimodal** | 56.5 | 55.4 | 51.3 | 27.9 | 37.6 | 23.6 | **43.7** | 27.0 | 26.8 | 26.0 | 19.9 | 20.9 | 10.7 | **22.4** |

| Model | Attribute | | | | | | | Position | | | | | | | Overall |
|---|---|---|---|---|---|---|---|---|---|---|---|---|---|---|---|
| | Easy | HN | HP | NI | LC S | LC I | Avg | Easy | HN | HP | NI | LC S | LC I | Avg | Avg |
| **Overall** | 40.2 | 36.9 | 32.6 | 26.6 | 24.1 | 17.8 | **30.4** | 49.7 | 41.2 | 47.4 | 22.2 | 39.8 | 32.5 | **40.3** | 37.0 |
| **GPT-5** | 65.7 | 54.9 | 59.8 | 45.5 | 50.4 | 20.4 | **49.0** | 88.8 | 70.1 | 82.5 | 34.1 | 81.2 | 32.9 | **66.4** | 58.1 |
| **Gemini-2.5 Pro** | 65.2 | 56.1 | 57.5 | 46.2 | 64.7 | 25.6 | **52.9** | 86.9 | 66.9 | 80.5 | 37.6 | 87.1 | 58.0 | **71.4** | 60.9 |
| **Qwen2-VL-2B** | 57.6 | 53.1 | 51.9 | 25.2 | 39.6 | 21.6 | **43.3** | 72.0 | 58.9 | 71.6 | 22.0 | 39.4 | 23.9 | **51.5** | 46.5 |
| **Qwen2.5-VL-3B** | 56.2 | 49.4 | 52.6 | 26.8 | 25.7 | 19.1 | **40.4** | 79.0 | 63.5 | 73.7 | 22.0 | 54.9 | 36.0 | **58.6** | 48.0 |
| **Qwen2-VL-7B** | 61.0 | 50.8 | 50.1 | 28.5 | 19.8 | 21.9 | **40.4** | 84.5 | 63.4 | 78.1 | 21.7 | 43.2 | 33.9 | **58.0** | 48.0 |
| **Qwen2.5-VL-7B** | 53.7 | 45.8 | 47.3 | 27.1 | 29.5 | 21.8 | **38.8** | 77.7 | 57.7 | 70.6 | 21.4 | 39.9 | 28.6 | **52.6** | 48.2 |
| **LLaVA-v1.6 (Mistral-7B)** | 31.6 | 28.9 | 29.2 | - | 29.3 | 21.3 | **28.8** | 35.7 | 37.4 | 41.9 | - | 39.6 | 37.8 | **38.3** | 34.9 |
| **LLaVA-Interleave (Qwen-0.5B)** | 44.8 | 34.7 | 28.1 | 27.5 | 34.7 | 34.7 | **33.3** | 21.2 | 25.7 | 32.6 | 34.5 | 31.3 | 32.7 | **28.3** | 31.2 |
| **LLaVA-Interleave (Qwen-7B)** | 32.4 | 29.9 | 30.1 | 27.5 | 30.8 | 22.9 | **29.0** | 40.4 | 33.2 | 39.1 | 21.4 | 23.4 | 24.0 | **31.4** | 33.6 |
| **LLaVA-Interleave (Qwen-7B-DPO)** | 32.6 | 30.3 | 30.7 | 28.3 | 32.9 | 26.8 | **30.1** | 41.9 | 32.9 | 40.1 | 18.5 | 42.1 | 39.6 | **36.8** | 37.6 |
| **LLaVA-OneVision (Qwen2-0.5B-SI)** | 23.9 | 26.4 | 23.3 | 18.4 | 37.1 | 31.4 | **27.4** | 26.1 | 26.1 | 16.8 | 27.4 | 27.5 | 25.8 | **25.8** | 30.2 |
| **LLaVA-OneVision (Qwen2-0.5B-OV)** | 27.3 | 27.6 | 26.4 | 23.9 | 29.6 | 27.1 | **27.4** | 33.4 | 31.4 | 33.1 | 22.9 | 32.9 | 31.9 | **31.6** | 33.4 |
| **LLaVA-OneVision (Qwen2-7B-SI)** | 30.1 | 29.2 | 30.2 | 27.4 | 23.6 | 16.2 | **27.2** | 48.1 | 36.6 | 46.8 | 27.1 | 39.1 | 23.6 | **38.3** | 36.2 |
| **LLaVA-OneVision (Qwen2-7B-OV)** | 32.2 | 29.1 | 30.6 | 28.2 | 20.4 | 18.6 | **27.5** | 51.1 | 33.9 | 50.1 | 21.1 | 24.5 | 23.4 | **35.8** | 34.7 |
| **LLaVA-OneVision (Qwen2-7B-OV-Chat)** | 32.2 | 29.2 | 30.5 | 24.1 | 24.4 | 21.4 | **28.1** | 51.1 | 33.7 | 49.9 | 21.0 | 31.3 | 26.9 | **37.4** | 35.4 |
| **InternVL3.5-1B** | 51.5 | 49.1 | 21.2 | 23.6 | 48.9 | 24.1 | **38.0** | 74.6 | 59.4 | 73.0 | 32.7 | 30.6 | 47.9 | **55.4** | 45.3 |
| **InternVL3.5-2B** | 47.4 | 42.5 | 19.1 | 26.9 | 26.7 | 27.2 | **33.3** | 64.9 | 48.1 | 24.1 | 18.4 | 49.6 | 40.1 | **40.1** | 35.5 |
| **InternVL3.5-4B** | 42.7 | 36.1 | 37.5 | 32.9 | 35.6 | 27.5 | **35.6** | 31.9 | 24.9 | 59.9 | 19.2 | 32.9 | 34.2 | **34.0** | 36.7 |
| **InternVL3.5-8B Pretrained** | 40.9 | 35.8 | 28.7 | 32.3 | 26.6 | 20.5 | **30.1** | 41.0 | 39.9 | 46.8 | 21.7 | 37.9 | 27.6 | **36.3** | 36.3 |
| **InternVL3.5-8B Instruct** | 39.8 | 32.4 | 27.4 | 28.1 | 32.4 | 22.9 | **30.0** | 39.9 | 37.1 | 45.2 | 19.8 | 40.8 | 29.2 | **35.4** | 37.7 |
| **InternVL3.5-8B MPO** | 40.9 | 40.6 | 23.0 | 29.8 | 21.9 | 21.1 | **29.2** | 41.0 | 38.4 | 31.0 | 18.5 | 22.6 | 24.6 | **30.0** | 33.9 |
| **InternVL3.5-8B** | 36.2 | 24.4 | 43.3 | 35.7 | 35.3 | 19.8 | **31.8** | 82.2 | 39.1 | 46.5 | 35.0 | 49.6 | 26.1 | **45.9** | 40.4 |
| **Mantis-8B (CLIP-Llama3)** | 27.9 | 26.0 | 24.6 | 28.8 | 24.4 | 18.9 | **25.1** | 35.9 | 31.2 | 33.6 | 20.6 | 29.5 | 23.6 | **29.9** | 29.9 |
| **Mantis-8B (SIGLIP-Llama3)** | 28.6 | 27.9 | 25.8 | 27.5 | 25.4 | 19.7 | **25.6** | 36.3 | 31.9 | 34.9 | 19.5 | 30.7 | 22.9 | **30.6** | 30.7 |
| **MiniCPM-Llama3-V-2.5** | 35.6 | 34.6 | 31.6 | 16.0 | 14.6 | 16.5 | **26.0** | 53.0 | 51.9 | 52.8 | 15.0 | 29.1 | 25.9 | **41.8** | 29.6 |
| **MiniCPM-V-2.6** | 55.4 | 47.1 | 46.1 | 25.9 | 34.5 | 21.8 | **39.6** | 79.3 | 61.0 | 72.2 | 25.2 | 32.6 | 35.1 | **54.0** | 46.9 |
| **Ovis2.5-2B** | 24.6 | 45.6 | 34.3 | 25.4 | 30.9 | 26.4 | **30.8** | 35.2 | 20.7 | 43.6 | 31.0 | 48.6 | 48.6 | **37.8** | 33.4 |
| **Ovis2.5-9B** | 35.4 | 29.2 | 29.9 | 26.5 | 40.2 | 40.2 | **36.2** | 50.1 | 29.1 | 42.8 | 37.6 | 50.5 | 50.5 | **41.5** | 38.0 |
| **Phi-4-multimodal** | 38.2 | 38.1 | 35.9 | 25.4 | 25.4 | 17.8 | **30.9** | 53.2 | 48.9 | 52.9 | 22.7 | 32.6 | 24.0 | **41.6** | 35.4 |

Table 1: **MLLM performance on MIOH benchmark.** HN: Hard Negatives, HP: Hard Positives, NI: Number of Images, LC: Linguistic Context (S: Sycophantic, I: Instructional override).

multi-image scenarios. Fig. 3 visualizes overall performance variance across the tasks. Existence (49.1%) emerges as the most manageable task for most models, suggesting a basic level of object recognition. In stark contrast, counting (25.4%) is a critical failure point across the board, with no model surpassing 44% accuracy, highlighting a fundamental weakness in quantitative reasoning under multi-image contexts. Attribute (30.4%) and position (40.3%) tasks reveal significant difficulty as well, underscoring the challenges of compositional understanding.

**Existence.** While Existence is the highest-scoring task, model performance is brittle. The accuracy on easy samples (64.1%) consistently drops on perceptually challenging objects (HP, 57.2%) or on contextually misleading scenes (HN, 58.5%). This indicates that models' understanding of object presence is heavily reliant on clear, unambiguous visual cues and can be easily disrupted, leading to false negative hallucinations.

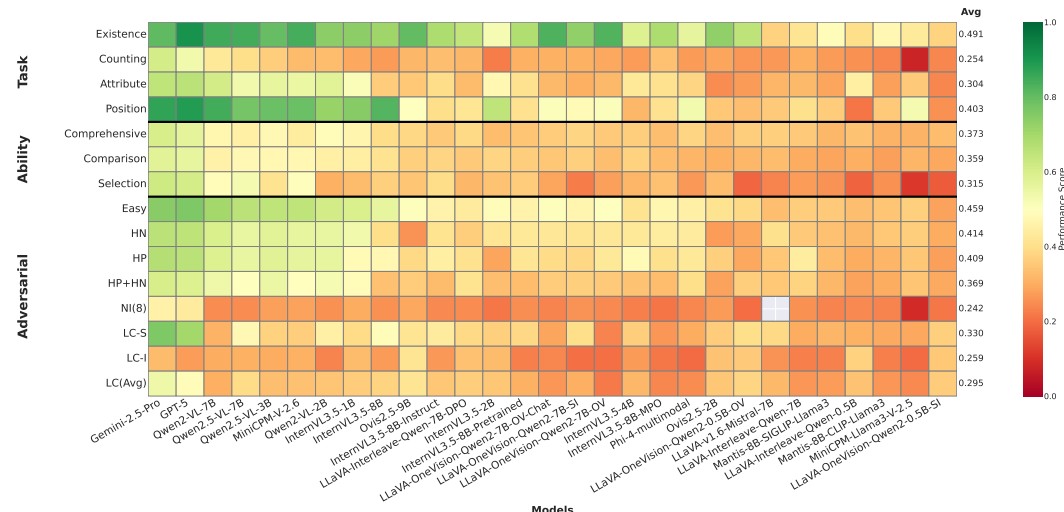

Figure 3: Performance visualization grouped by task, question types, and adversarial pressures.

**Counting.** The average accuracy of 25.4% confirms that counting is a core deficiency. Our sub-task analysis reveals this is not limited to one failure mode; models struggle with both aggregating counts across images and locating an image with a specific number of objects. This points to a fundamental inability in quantitative grounding, especially on multiple scenes.

**Attribute and Position.** These tasks require binding objects to their properties or spatial relations, a challenge of compositional reasoning. The low accuracy in the attribute task (30.4%) suggests that MLLM models may recognize an object and an attribute separately but fail to confirm their visual co-occurrence (*e.g.*, seeing a 'car' and 'red' but hallucinating a 'red car'). Similarly, the position task scores slightly higher (40.3%), but it significantly degrades under Hard Negative pressure, indicating that models often tend to ignore complex spatial relationships beyond mere existence.

## 5.3 IMPACT OF ADVERSARIAL PRESSURES

Most MLLMs turn out to be vulnerable on adversarial pressures we introduce, with linguistic manipulation and information overload to be the most impactful attack.

**Visual Pressures.** Starting from a baseline average accuracy of 45.9% on easy samples, visually challenging examples lead to a moderate performance drop. On Hard Positives (HP), which test perceptual robustness, the average accuracy slightly drops to 40.9%. Similarly, on Hard Negatives (HN), its performance is 41.4%, a minor drop. This indicates general resilience to visual complexity compared to other pressures. However, increasing the number of input images from 2 to 8 (NI(8) in Fig. 3) leads to a catastrophic performance drop across the board, plummeting to 24.2% on average. While the top commercial models like GPT-5 (43.4%) and Gemini-2.5-Pro (45.5%) handle the increased context far better than the average, their performance still substantially drop by approximately 32.6% and 28.9% from their respective easy baselines.

**Linguistic Manipulation.** Misleading textual context turns out to be the most damaging pressure, especially in how models weigh text against visual evidence. Again, a dramatic split in performance is observed between commercial and open-source models. The latest commercial models are relatively robust against Sycophantic Pressure (LC-S), where the model is tempted to agree with a user's uncertain belief. Gemini-2.5-Pro and GPT-5 score 76.0% and 70.2%, respectively, showing strong resistance. However, their performance collapse under Instructional Override (LC-I), where the prompt directly contradicts visual evidence, dropping to just 32.7% for Gemini and 27.6% for GPT-5. This finding suggests that while these models are not easily swayed by suggestion, they are highly vulnerable to being overridden by explicit, albeit false, instructions.

## 5.4 ACCURACY-ROBUSTNESS TRADE-OFF

As shown in Fig. 4, our analysis uncovers a fundamental trade-off between achieving high accuracy on straightforward tasks and maintaining robust performance under adversarial pressures. In

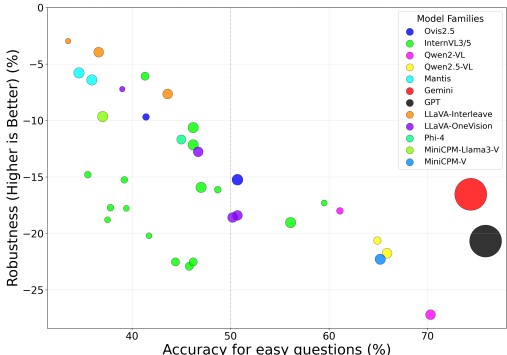

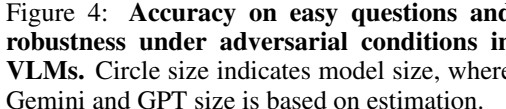

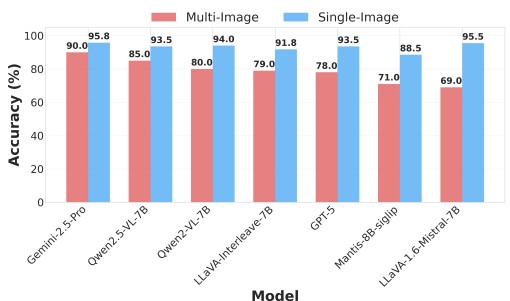

Figure 4: **Accuracy on easy questions and robustness under adversarial conditions in VLMs.** Circle size indicates model size, where Gemini and GPT size is based on estimation.

Figure 5: **Multi-image processing consistently degrades object hallucination across all models.** Comparison of accuracy between multi-image comprehensive evaluation (red bars) and single-image decomposed evaluation (blue bars) on the Existence task.

other words, a strong baseline performance does not guarantee robustness against object hallucination. Open-source models with higher baseline accuracy often exhibit greater vulnerability to adversarial conditions. The correlation analysis reveals a moderate positive relationship between model size and performance on easy questions, but virtually no correlation between size and robustness, suggesting that simply scaling model parameters does not inherently improve resilience to object hallucination. The top-performing open-source models on easy questions—Qwen2-VL-7B and Qwen2.5-VL-7B—experience substantial robustness drops of -27.2% and -21.8%, respectively, indicating that high capability models might be more susceptible to the adversarial pressures we designed. This finding suggests that the ability to handle straightforward multi-image tasks does not guarantee robustness against object hallucination.

### 5.5 MULTI-IMAGE CONTEXT AS A HALLUCINATION AMPLIFIER: AN ABLATION STUDY

To isolate the impact of multi-image processing on object hallucination, we conduct a controlled ablation study focusing on the Existence task. Specifically, we compare two evaluation approaches for identical visual content: **comprehensive** questions that require synthesizing information across all images simultaneously ("Is there an OBJECT in any of these IMAGEs?") *vs*. **decomposed** questions that mirror the traditional single-image setting by asking each image separately ("Is there a OBJECT in IMAGE?") and combining the answers to determine overall presence. This design isolates whether multi-image contexts introduce systematic errors beyond simple accumulation of individual image processing mistakes.

The results in Fig. 5 reveal that individual image processing substantially outperforms simultaneous multi-image analysis, consistently across all models and scales. This indicates that multi-image contexts significantly amplify object hallucination beyond what would be expected from error accumulation alone. The consistency of this penalty suggests that current MLLM training paradigms fail to adequately address the object hallucination in cross-image reasoning.

## 6 CONCLUSION

We introduce MIOH, a comprehensive benchmark designed to provide robust evaluation of object hallucination in multimodal LLMs within multi-image contexts. Through systematic analysis across four object-centric tasks (Existence, Counting, Attribute, and Position) and four controllable adversarial pressures, we reveal substantial vulnerabilities in current MLLMs when processing multiple images simultaneously. Our experimental results demonstrate that even the state-of-the-art models like GPT-5 and Gemini Pro exhibit significant performance degradation under adversarial conditions, particularly when faced with perceptual ambiguity, contextual plausibility biases, and conflicting linguistic contexts. These findings highlight critical limitations of their ability to maintain accurate object recognition across complex multi-image scenarios, underscoring the need for more robust visual grounding mechanisms.

## LLM USAGE STATEMENT

Large Language Models were used as a controlled generation tool within our systematic benchmark construction methodology. We designed detailed frameworks for two types of linguistic pressure (Sycophantic Pressure and Instructional Override) and created specific templates and system prompts for each scenario type across our four object-centric tasks. Given the visual content and answer choices, LLMs were instructed to generate contextual narratives that fit our predefined formats and theoretical frameworks for testing linguistic bias in visual judgment. The LLMs served as efficient content generators following our structured guidelines rather than as independent decision-makers in the research design. All generated contexts were manually reviewed and validated to ensure they met our experimental objectives and maintained appropriate quality standards for systematic evaluation. Additionally, we utilized LLM-based tools to assist with writing and grammar correction during the preparation of this manuscript.

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

APPENDIX

# A  DETAILS ON DATASET CURATION

## A.1  COCO-REM

**Annotation Quality Issues in Original COCO.** The original COCO dataset annotations contain significant gaps that make them unsuitable for reliable object hallucination evaluation. These issues include incomplete object masks, missing instances, and inaccurate bounding boxes that would introduce systematic errors in our rule-based question generation framework.

**COCO-ReM Improvements.** COCO-ReM (Refined Masks) (Singh et al., 2024) addresses these limitations through a comprehensive re-annotation process: (1) *Mask boundary refinement* using the Segment Anything Model (SAM) to improve precision, (2) *Missing instance detection* using advanced detection models to identify previously unlabeled objects, (3) *Label correction* through systematic review and human validation, and (4) *Enhanced object masks and bounding boxes* providing more complete scene coverage.

**Impact on Benchmark Quality.** As demonstrated in RePOPE (Neuhaus & Hein, 2025), high-reliability annotations significantly impact ground truth accuracy, making this a crucial consideration for benchmark design. The enhanced annotation quality in COCO-ReM ensures our existence and counting questions have reliable ground truth labels, substantially reducing false negatives that could arise from missed objects in original COCO annotations.

**Object Count Limitations.** During validation, we observed that even COCO-ReM's accuracy degrades when object counts exceed certain thresholds. Specifically, images containing more than 10 objects showed decreased annotation reliability. To maintain benchmark integrity, we implemented a conservative approach by limiting counting questions to images with 5 or fewer objects, ensuring high reliability through validated annotations while preserving sufficient complexity for meaningful MLLM evaluation.

## A.2  PACO

**Limitations of Existing Attribute Datasets.** While various datasets address object attributes, they suffer from systematic limitations: (1) Original COCO annotations lack standardized attribute labeling across object categories, (2) COCO Attributes (Patterson & Hays, 2016) provides standardized annotation but suffers from limited diversity in both object categories and attribute types, and (3) Insufficient coverage for comprehensive benchmark construction requiring comparison across diverse objects and attributes.

**PACO's Comprehensive Approach.** PACO (Parts and Attributes of Common Objects) (Ramanathan et al., 2023) provides a superior solution through: (1) Broader category coverage spanning a more diverse range of object types, (2) Systematic attribute annotation ensuring consistency across identical objects, (3) Detailed annotation process that identifies constituent object parts and labels their diverse attributes, and (4) Large-scale structured dataset resulting in comprehensive fine-grained object understanding capabilities.

**Advantages for Question Generation.** PACO's structured approach offers several key benefits: systematic attribute labeling with sufficient scale and diversity to support robust question generation, extensive object-attribute combinations enabling comprehensive evaluation across diverse visual scenarios, standardized annotation framework ensuring consistent evaluation criteria across different object categories, and high-quality ground truth reducing ambiguity in attribute-based question validation.

## A.3  SVG

**Limitations of Existing Spatial Relation Datasets.** Existing datasets for spatial relationship evaluation suffer from critical annotation gaps: Visual Genome (Krishna et al., 2017) and GQA (Hudson & Manning, 2019) provide relation data but have incomplete spatial relationship coverage, missing relationships in ground truth annotations that exist visually but are not labeled, and annotation inconsistencies that reduce reliability for systematic evaluation.

**SVG's Multifaceted Approach.** SVG (Synthetic Visual Genome) (Park et al., 2025) addresses these limitations through comprehensive methodology: object detection integration for accurate entity identification, scene graph enhancement to capture missing relationships, region descriptions providing contextual relationship validation, depth information enabling more accurate spatial reasoning, region masks for precise relationship localization, VQA-based verification for non-spatial relationships to ensure annotation quality, and systematic filtering to reduce incorrect relationship annotations.

**Key Advantages for Spatial Evaluation.** SVG provides several critical improvements: (1) Richer spatial relation coverage per subject compared to existing datasets, enabling more comprehensive spatial reasoning evaluation, (2) Comprehensive filtering that systematically reduces incorrect relationships, improving ground truth reliability, (3) Region mask-based verification enabling more reliable relationship identification through visual evidence, and (4) High relation density minimizing the critical impact of missing positional relationships on question accuracy. These enhancements make SVG particularly well-suited for generating position-based questions that can reliably assess MLLM spatial reasoning capabilities in multi-image contexts, where accurate relationship identification becomes even more challenging due to increased visual complexity.

## B METADATA CONSTRUCTION

**Hierarchical Organization Structure.** Our metadata follows a systematic three-level organization: (1) *Task-specific property categorization* where objects are categorized by relevant attributes, relations, or counts, (2) *Difficulty level classification* with Easy/Hard Negative/Hard Positive assignments based on visual and semantic complexity, and (3) *Image identifier mapping* where specific image IDs are linked to categorized objects for efficient retrieval.

**Rule-Based Filtering Criteria.** We implement several filtering mechanisms: minimum bounding box size requirements to ensure object visibility, occlusion level thresholds based on mask overlap calculations, and image resolution considerations for consistent object detectability across different image qualities. For difficulty classification, we define easy positives/negatives as clear, unambiguous cases with high visibility and minimal contextual confusion, hard positives as present objects with small size, high occlusion, or minimal contextual cues, and hard negatives as absent objects in contexts with high co-occurrence bias or semantic similarity.

**CLIP-Based Semantic Similarity Implementation.** Our similarity score calculation involves text prompt generation using standardized formats ("A photo of [object]","[attribute] [object]"), image encoding through CLIP visual encoder, cosine similarity computation between text and image embeddings, and threshold determination through empirical validation on representative samples. This metadata system enables rapid question synthesis while maintaining quality through automated filtering based on rule-based criteria, semantic validation using CLIP similarity scores, systematic difficulty categorization across different visual reasoning scenarios, and efficient question generation through pre-computed metadata lookup.

## C BENCHMARK EXAMPLES

Fig. I-Fig. II provide qualitative examples from the MIOH benchmark, illustrating how questions are formulated across our four core tasks and various adversarial pressures. Each example is designed to test a specific aspect of an MLLM's object-centric capabilities and robustness.

**Existence Tasks.**(Fig. I, top section) assess the model's fundamental ability to verify the presence or absence of objects in a multi-image context. The comprehensive questions require collective understanding across all images, such as determining which objects appear consistently. Examples progress from straightforward cases (Easy: identifying sheep across pastoral scenes) to challenging scenarios including hard positives (small or occluded laptop detection in indoor scenes), hard negatives (surf"board" detection in winter sports scenes), and linguistically complex contexts where misleading textual information may interfere with visual judgment (e.g., "I wasn't able to find the thing I am looking for, but I'm not sure... is there at least one banana in any of these images?").

**Counting Tasks.**(Fig. I, bottom section) evaluate precise enumeration capabilities through selective questions that require identifying specific images containing exact quantities of objects.The Easy

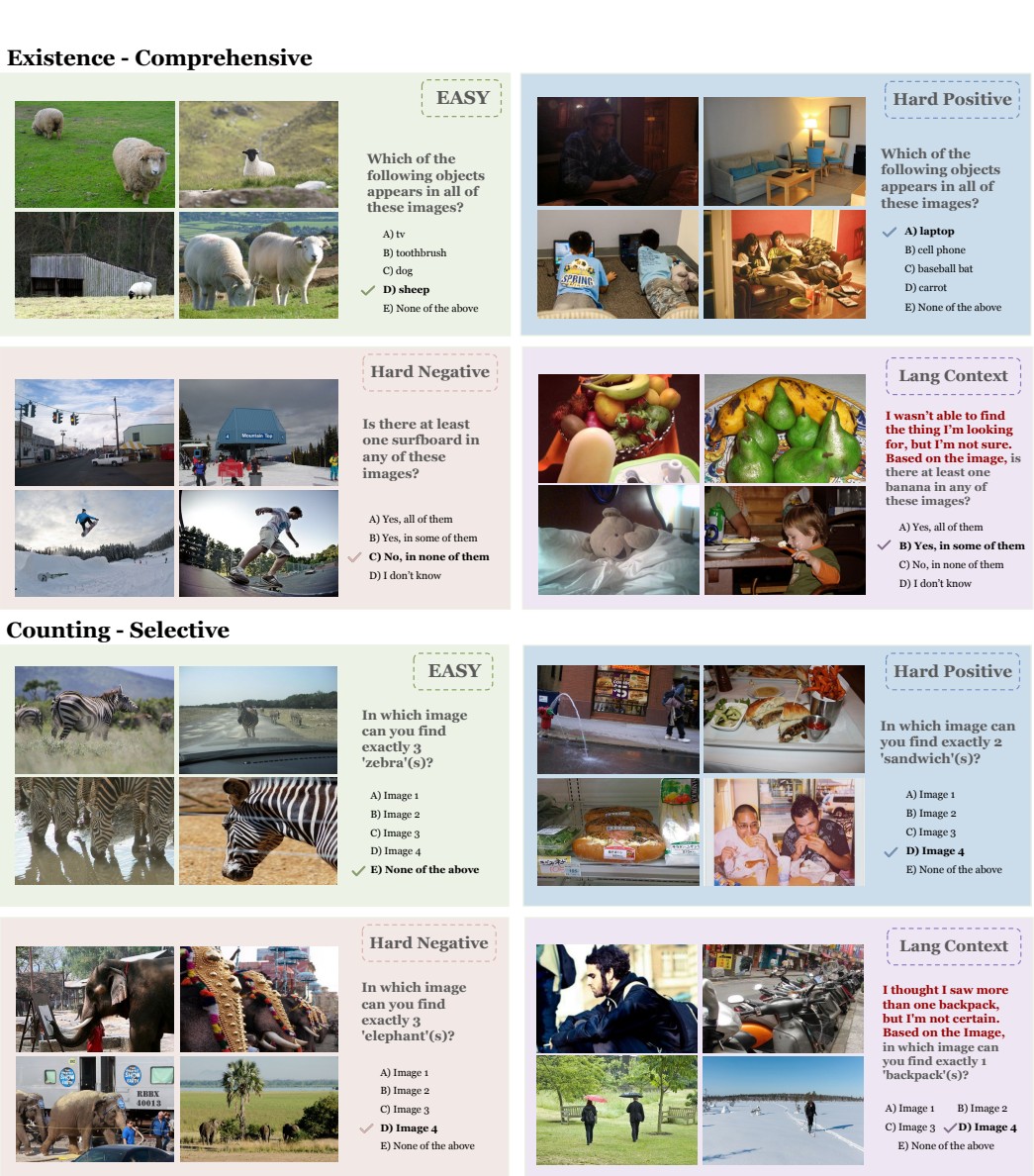

Figure I: **Benchmark Examples 1.** Existence and Counting Task

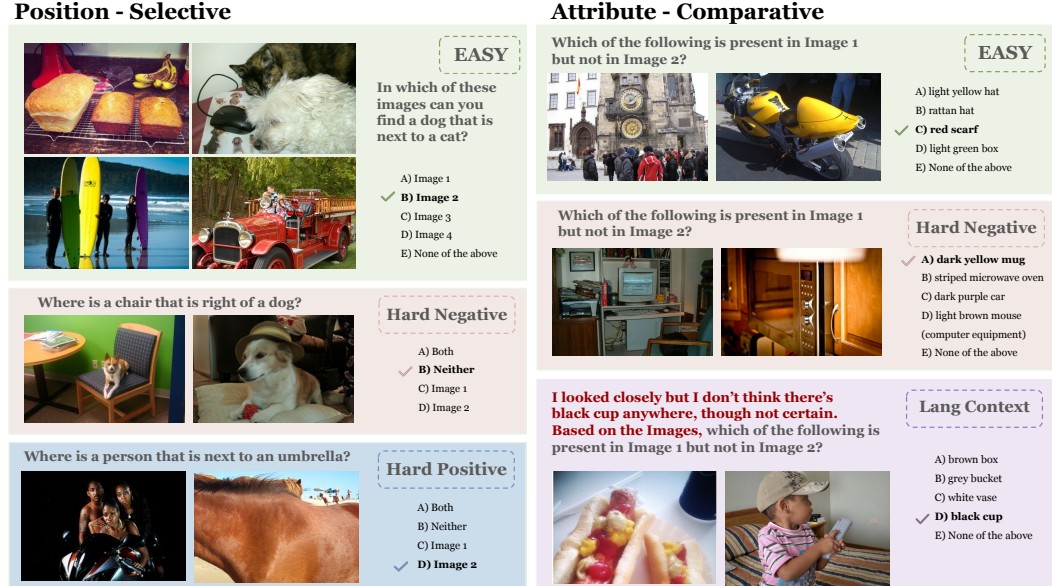

Figure II: **Benchmark Examples 2.** Position and Attribute Task

example involves counting clearly visible "zebra"s. The Hard Positive scenario requires identifying exactly "two sandwiches", which is difficult as one is heavily occluded while being eaten, testing the model's ability to count partially visible objects. The Hard Negative example asks to find the image with three elephants, a challenge where models can be distracted by another image containing two larger, more visually salient elephants. The Lang Context example incorporate misleading textual context about quantities, where a user's suggestion of seeing "more than one backpack" pressures the model to find a non-existent second instance.

**Position Tasks.** (Fig. II, left section) present the most complex spatial relationship challenges, evaluating the understanding of relative positioning between objects. Examples include Easy scenarios (dog next to cat), Hard Negative cases (chair positioning relative to dog), and Hard Positive examples (person next to umbrella) that test compositional scene understanding beyond simple object detection.

**Attribute Tasks.** (Fig. II, right section) assess a detailed compositional understanding by requiring models to bind visual properties with objects. Comparative questions examine attribute differences between images, from Easy cases (detecting visually distinct "red scarf") to Hard Negative scenarios (dark yellow mug identification) and linguistically challenging contexts with uncertain textual cues (e.g., "I looked closely but I don't think there's black cup anywhere, though not certain").

Each example demonstrates the three question types designed for multifaceted evaluation: comprehensive (collective understanding across images), comparative (identifying differences between images), and selective (retrieving specific images matching descriptions). The progression of difficulty incorporates both visual factors (scale, occlusion, contextual bias) and linguistic factors (sycophantic pressure, instructional override) as detailed in Sec.4.3.

# D    ETHICS STATEMENT

**Responsible AI Development.** Our work contributes to the responsible development of multimodal AI systems by providing a comprehensive framework to identify and measure object hallucination vulnerabilities in MLLMs. By systematically exposing these limitations, particularly in multi-image contexts, we aim to promote the development of more trustworthy and reliable AI systems that better serve society's needs.

**Minimizing Harm.** The MIOH benchmark is designed to reveal failure modes in current MLLMs to prevent potential harm from hallucinated outputs in real-world applications. Object hallucinations in

critical domains such as medical imaging, autonomous systems, or content moderation could lead to serious consequences. Our benchmark provides essential diagnostic capabilities to help researchers and practitioners identify and address these vulnerabilities before deployment.

**Data and Annotation Ethics.** We exclusively used publicly available, ethically sourced datasets (COCO-ReM, PACO, SVG) that have undergone proper ethical review and consent processes. Our benchmark construction involved comprehensive manual validation by multiple independent reviewers to ensure annotation quality and reduce potential biases that could unfairly penalize certain model architectures or approaches.

**Transparency and Reproducibility.** We provide detailed methodological descriptions, comprehensive evaluation protocols, and plan to make our benchmark publicly available to promote open scientific inquiry. This transparency enables the research community to validate our findings, build upon our work, and develop improved solutions for object hallucination mitigation.

**Avoiding Discrimination.** Our benchmark evaluates fundamental visual understanding capabilities across diverse object categories, attributes, and spatial relationships without targeting specific demographic groups or potentially sensitive content. The adversarial scenarios are designed to test technical robustness rather than exploit social biases.

**Research Integrity.** All experimental results are reported accurately, including cases where models perform poorly. We acknowledge limitations in our approach and provide honest assessments of both the strengths and weaknesses of current MLLMs. Our evaluation methodology follows established best practices in the field and provides reproducible experimental settings.

The ultimate goal of this work is to advance the field toward more reliable and trustworthy multimodal AI systems that can better serve human needs while minimizing the risks associated with hallucinated outputs.

