# OpenReview forum: "Towards Robust Benchmark of Object Hallucination on Multiple Images"
_ICLR.cc/2026/Conference — ICLR 2026 Conference Withdrawn Submission_

### Official Review · Reviewer_qnit · 2025-10-25

**Soundness:** 2
**Presentation:** 2
**Contribution:** 2
**Rating:** 2
**Confidence:** 4

**Summary:**

This paper introduces the Multi-Image Object Hallucination (MIOH) benchmark, created to address key limitations in current evaluation methods. The authors argue that existing benchmarks are often confined to single-image settings or provide only superficial, high-level assessments for multi-image contexts. As a result, they fail to diagnose the nuanced causes behind object hallucination. MIOH overcomes this by systematically integrating four fundamental object-centric tasks—existence, counting, attribute, and position—with four controllable adversarial factors, such as visual context scale and misleading textual cues.

**Strengths:**

1. The paper addresses a timely and significant problem. Multi-image reasoning is a critical and rapidly developing capability for MLLMs, and robustly evaluating it is essential, particularly as models are increasingly applied to sequential visual data like videos.

2. The paper is supported by comprehensive experiments covering a wide spectrum of models, from leading proprietary systems to various open-source alternatives.

3. MIOH thoughtfully incorporates controllable factors, including multi-task categories and, importantly, a suite of adversarial pressures. This allows for a much more diagnostic evaluation than a simple accuracy score.

**Weaknesses:**

1. The paper's central claim to novelty is questionable, as it largely ignores the extensive work on video hallucination benchmarks like Video-Halluci, VidHalluc, and even the multi-image components of HallusionBench. Since video is inherently a multi-image problem, these frameworks already assess object consistency and factual accuracy across frames. The paper fails to articulate what unique conclusions MIOH can provide that are not already captured by these more complex and established video-centric evaluations.

2. Furthermore, the specific adversarial factors are not original and have been thoroughly explored in recent work. For instance, the impact of linguistic priors and visual confusion has been a core focus of benchmarks like PhD (CVPR 2025) and VLind-bench (NAACL), while the task categories themselves mirror those found in AMBER. When the multi-image aspect is viewed as a subset of video, the benchmark appears highly derivative, borrowing its core testing principles from these existing works without significant innovation.

3. Methodologically, the benchmark feels more like a compilation of existing ideas than a new task. The use of multiple-choice questions is a standard evaluation format that offers no clear advantage over other common methods like binary assessment. As a result, the overall design comes across as an aggregation of established datasets and evaluation paradigms, rather than a fundamentally new contribution to the field.

**Questions:**

See the weaknesses.

The paper needs to clarify its key distinctions from similar frameworks, especially video benchmarks. Specifically, what unique conclusions does it offer that others cannot, and what are its unique design elements that enable these insights? Furthermore, the authors should provide more detailed, actionable guidance on how the benchmark can direct future MLLM improvements—for instance, by explicitly linking poor performance on specific categories to concrete suggestions for model refinement.

---

### Official Review · Reviewer_8QJc · 2025-10-25

**Soundness:** 2
**Presentation:** 3
**Contribution:** 2
**Rating:** 2
**Confidence:** 3

**Summary:**

The authors propose the MIOH benchmark to evaluate multi-image object hallucination which contains  4 object-centric tasks (existence, counting, attribute, position) via 3 question types (comprehensive, comparative, selective).

**Strengths:**

Evaluations on 30 models (including GPT-5 and Gemini-2.5-Pro) show overall average accuracy of only 37.0%, with SOTA models still suffering significant performance drops under adversarial conditions; multi-image context amplifies hallucination, counting is the most challenging task, and misleading text/ increased image count (≥8) have the strongest negative impacts.

**Weaknesses:**

1. While multi-image tasks hold significance, multi-image problems that lack temporal scenarios have extremely limited practical relevance. Specifically, it fails to evaluate MLLMs’ ability to track object trajectories or reason about temporal dependencies—capabilities critical for real-world applications such as autonomous driving and video surveillance.
2. This very limitation, I suspect, explains why the authors were only able to restrict their evaluation tasks to Visual Question Answering (VAQ) problems, which itself also suffers from inherent limitations. Furthermore, such evaluations cannot reflect which specific capabilities of a model have improved when it achieves better performance after undergoing training (e.g., reinforcement learning, RL). Simple multiple-choice questions, devoid of reasoning verification, are highly vulnerable to being "hacked." This is precisely why I personally argue that current benchmark efforts in the field generally lack the significance to guide model development.
3. The benchmark uses "accuracy" as the sole metric, which only judges whether hallucination occurs but cannot quantify hallucination severity (e.g., distinguishing "mislabeling a cat as a dog" from "inventing a non-existent tiger"), failing to reflect risk gradients in practical use.
4. The paper does not investigate whether hallucination rises due to visual token overload (e.g., models dropping critical visual tokens when processing >8 images, as hinted at in Sec. 5.3 but not verified).

**Questions:**

Q1: You limited counting tasks to images with ≤5 objects (due to reduced annotation reliability of COCO-ReM for >10 objects), but real-world multi-image scenarios (e.g., crowded streets, busy warehouses) often involve more than 5 objects. Have you explored alternative datasets (e.g., synthetic datasets with verified counts for 6–15 objects) or annotation strategies (e.g., combining COCO-ReM with crowd-sourced validation for high-object-density images) to evaluate model counting capabilities in such practical scenarios? If not, do you believe the current ≤5-object limit underestimates how models struggle with quantitative reasoning in dense multi-image contexts?

Q2: Your results show that performance degrades in multi-image scenarios. Could you provide a simple analysis explaining how to improve the multi-image performance of open-source models, or how to bridge the current gap between open-source and closed-source models in this aspect?

---

### Official Review · Reviewer_pbQS · 2025-10-29

**Soundness:** 3
**Presentation:** 3
**Contribution:** 2
**Rating:** 4
**Confidence:** 4

**Summary:**

The paper introduces MIOH, a diagnostic benchmark for object hallucination specifically under multi-image inputs. It combines four object-centric tasks (existence, counting, attribute, position) with multi-image query forms and adds controllable adversarial factors to probe failure modes; the experiments show sizable drops from normal to adversarial settings and that processing images jointly amplifies hallucinations beyond simple per-image error accumulation.

**Strengths:**

1. Writing is polished and easy to follow, with a clean structure from benchmark design to analyses and figures that reinforce the main claims。
2. Evaluation on many models, enabling credible cross-model comparisons and trend analysis across both open-source and frontier proprietary systems.

**Weaknesses:**

1. The paper explicitly claims “the first comprehensive evaluation framework specifically designed to assess object hallucination under multi-image settings.” This claim is not accurate. Prior work has already released benchmarks that target multi-image hallucination, eg. MIHBench[1] which is designed to evaluate hallucination in multi-image MLLMs and provides a systematic study with tasks for existence, count, and identity consistency, along with a mitigation method.
2. Heavy reliance on template-based multiple-choice evaluation limits ecological validity, and robustness to paraphrase or option order is not reported.
3. No mitigation method is proposed to address the identified hallucination failures

[1]: MIHBench: Benchmarking and Mitigating Multi-Image Hallucinations in Multimodal Large Language Models. ACM MM25.

**Questions:**

1. Do your conclusions persist when templates are paraphrased and when the images or answer-option order is shuffled; in other words, how sensitive are results to prompt wording and choice order.
2. Do you have any method that reduces these hallucinations under your adversarial factors?

---

### Official Review · Reviewer_hWgF · 2025-10-31

**Soundness:** 3
**Presentation:** 3
**Contribution:** 2
**Rating:** 6
**Confidence:** 4

**Summary:**

This paper introduces the MIOH benchmark to study object hallucination in multimodal large language models when dealing with multiple images. It focuses on four object-based tasks and four types of adversarial factors that can cause hallucinations. The study finds that even strong models like GPT-5 and Gemini Pro still make mistakes when the visual or text context becomes complex. The benchmark helps show how these models fail and can be used to improve their reliability in multi-image settings.

**Strengths:**

Strengths:
1. The paper introduces a clear and systematic benchmark (MIOH) that fills the gap in studying object hallucination under multi-image settings, combining key object tasks like existence, counting, attribute, and position.
2. It uses controllable adversarial factors to deeply analyze model weaknesses and provide fine-grained diagnostic insights.
3. The writing is clear and easy to understand, making the ideas accessible.
4. The experiments are thorough and cover most major multimodal models, showing strong empirical support for the conclusions.

**Weaknesses:**

Weaknesses:
1. While MIOH is comprehensive, it mainly focuses on object-level hallucination and may not cover higher-level reasoning or relational hallucinations between objects.
2. The analysis could go deeper on why certain models fail — for example, by linking errors to architecture or training differences.

**Questions:**

Questions:
1. Could the paper include some examples that better highlight multi-image object reasoning, such as relational or causal cases, to show how current models handle more complex reasoning scenarios?
2. Could the paper briefly analyze why models fail in some cases, just to give a bit more insight into the causes?

---

### Note · Authors · 2025-11-12

**Comment:**

After carefully considering the reviewers’ feedback and discussing within the author team, we agree with the reviewers’ comments and suggestions, and have decided to withdraw the paper in order to substantially revise and extend the work. We sincerely appreciate the reviewers’ constructive feedback and the committee’s time and effort, and we plan to resubmit a significantly improved version in the future.

**Withdrawal Confirmation:**

I have read and agree with the venue's withdrawal policy on behalf of myself and my co-authors.